# Productive Performance and Egg and Meat Quality of Two Indigenous Poultry Breeds in Vietnam, Ho and Dong Tao, Fed on Commercial Feed

**DOI:** 10.3390/ani10030408

**Published:** 2020-03-01

**Authors:** Duy Nguyen Van, Nassim Moula, Evelyne Moyse, Luc Do Duc, Ton Vu Dinh, Frederic Farnir

**Affiliations:** 1Centre for Interdisciplinary Research on Rural Development, Vietnam National University of Agriculture, Trau Quy, Gia Lam, Hanoi 100803, Vietnam; nvduy.hua@gmail.com; 2Fundamental and Applied Research in Animal and Health (FARAH) Department of Veterinary Management of Animal Resources, Faculty of Veterinary Medicine, University of Liege, Liege 4000, Belgium; nassim.moula@uliege.be (N.M.); evelyne.moyse@uliege.be (E.M.); 3Faculty of Animal Science, Vietnam National University of Agriculture, Trau Quy, Gia Lam, Hanoi 100803, Vietnam; dtghn@yahoo.co.uk

**Keywords:** indigenous chicken, body weight, meat, egg, production performances

## Abstract

**Simple Summary:**

Indigenous chicken breeds adapt well to the local conditions and provide the genetic diversity required to improve the development of poultry breeds. Nowadays, the intensive systems of chicken production use only hybrid lines with high genetic potential. These lines are a serious threat to the conservation of indigenous chicken breeds in the world. Therefore, research on indigenous chicken breeds are necessary in elucidating conservation and sustainable development strategies with respect to these chicken breeds. This work evaluates the production and laying performances, and the meat and egg quality of two breeds of Vietnamese broiler chickens, Ho and Dong Tao. Our work shows that the production performances of the two breeds are low compared to commercial lines. Improving the production and reproduction performances of these animals is necessary in contributing to the program of conservation and sustainable exploitation of these two emblematic breeds.

**Abstract:**

The objective of this work was the evaluation of the meat production and laying performances, and the meat and egg quality of two breeds of Vietnamese broiler chickens, Ho and Dong Tao, fed on a commercial diet. In a survey, we continuously recorded for 28 weeks, the data on the production performance and meat quality of 250 chicks from each breed. We investigated egg laying and egg quality using 36 Ho and 32 Dong Tao hens during 52 weeks of laying. The growth patterns were similar for the two breeds. Feed conversion ratios were also similar, and demonstrated the low efficiency of these two breeds when compared to commercial broilers. Slaughter age proved to affect several carcass yield characteristics, showing that slaughtering between 16 and 20 weeks might be better than at the usual age of 28 weeks. Yield, carcass composition and meat quality differed between the two studied breeds. The eggs production and number of embryonated eggs were low for the two breeds when compared to other breeds, with a lower hatching performance in Ho than in Dong Tao. In summary, the production performances of Ho and Dong Tao chickens were low, even when birds were fed a commercial diet. The study demonstrates the need to find ways to improve the production and reproduction performances of these animals, in order to contribute to the program of conservation and exploitation of these two breeds.

## 1. Introduction

Domestication of poultry started about 8000 years ago. Since then, humans have depended a lot on it. Chickens can adapt easily to various environmental conditions due to their small size, reduced needs and ability to find feed and water for themselves. Moreover, their short production cycle is another important advantage, making them a major source of animal protein for humans and providing diverse products for human consumption [1]. At the local scale, indigenous chicken populations have a significant contribution to household production, especially for low income farmers in Asia, Africa and the South Pacific [2].

Nowadays, because of improvements in the intensive farming systems, the poultry production has developed rapidly worldwide. Intensive poultry farming provides protein to the human population at a very large scale, especially in growing urban areas [3]. However, in the long term, this intensive poultry farming, using only high-yielding hybrid lines, is a serious threat to the genetic diversity of poultry breeds in the world because of the replacement of local breeds by these productive exotic breeds. According to an FAO report [4], 60 of the 1729 breeds identified worldwide are extinct, 154 are critically endangered, 214 are endangered and 1089 have an unknown conservation status. This trend could be particularly damaging for small-scale poultry breeders or livestock farmers in poor countries [3]. The rapid development of intensive poultry companies, not only leads to reductions in the number of indigenous chicken breeds, but also increases the dependence on commercial chicken lines production [5].

Intensively raised chicken breeds could show weaknesses in their ability to adapt to global warming, to emerging diseases and to complex changes in consumers demand. In that context, indigenous chicken breeds could be a rich genetic resource, able to provide solutions to problems eventually arising in selected chicken lines [6].

Vietnam is one of the chicken domestication centres and still has a rich genetic diversity of poultry breeds [7,8]. Vietnam has 14 indigenous chicken breeds, including Ri, Te, Tau Vang, Ac, Oke, H’mong, Tre, Choi, Phu Luu Te, To, Dan Khao, Mia, Ho and Dong Tao breed [9]. Another report from FAO even states that 28 indigenous chicken breeds are present in this country, but without listing these breeds [10]. For all these breeds, 7.3 million households use the backyard chicken farming system. This represents 92% of the households raising chicken. In this system, households usually keep 5–50 heads per household [11]. Industrial broilers represent 23.1% of the total Vietnamese chicken population. The remaining 76.9% is made up of native chickens or chickens obtained by crossing local and exotic chickens. For laying hens, the proportions are 43.3%, and 56.7%, respectively [12]. In local breeds of broilers, chickens are slaughtered at around five to six months of age, except for a small proportion of hens that are kept for laying until two years of age. Another report estimates that 84%-85% of the households in the Northeast and Northwest of Vietnam use that system [11]. The income from poultry production accounts for 32.5% of the total household income. In addition to their economic role, indigenous chicken breeds play an important role in the cultural and social relations of the Vietnamese people [9]. Poultry products (usually from indigenous chicken breeds) are used as gifts or offerings in important religious ceremonies [11].

The Ho and the Dong Tao breeds are two of the most important indigenous broiler chicken breeds in Vietnam. The population of Ho chickens is very small, with only 1404 individuals distributed in 88 households [13]. On the other hand, there are more than 1000 Dong Tao chicken households, totalizing nearly 10,000 individuals. Of these herds, only about 20 farms have a flock size exceeding 50 hens per household [14]. The price of Dong Tao and Ho chickens, comprised between 14 and 20 USD per kg, is more than 3-4 times the price of other Vietnamese chickens. Historically, the meat from these two breeds was offered as a gift to the King [10]. The names of the Ho and the Dong Tao breeds come from the region of origin of these breeds. The Ho breed is characterized by a large body size, with feathers of five varying colours (two for males and three for females, see Figure 1) and low reproductive performance [15]. The Dong Tao breed has completely different characteristics from the other indigenous chicken breeds from Vietnam. It has big sized legs, with feathers of five varying colours (two for males and three for females, see Figure 2) and comparatively higher body weight than the other Vietnamese indigenous chicken breeds [16]. The conservation of the Ho and the Dong Tao chicken breeds, as well as many other indigenous chicken breeds in Vietnam, is mainly achieved by the private sector in the farmers’ households. The aim of this study is to characterize the growth, laying, meat and eggs quality of these two emblematic chicken breeds of Vietnam when fed with a commercial diet. We also aim to compare these performances to other chicken breeds, in order to identify traits for which improvements are needed for an economically sustainable exploitation and conservation of these two breeds.

## 2. Materials and Methods

This study was carried out from June 2016 to July 2017. It involved simultaneously the two breeds, and was performed at the Faculty of Animal Science of the Vietnam National University of Agriculture in Hanoi (Vietnam). Hanoi is located at the latitude of 21°0278′ north and at the longitude of 105° 8432′ east. The weather is sub-tropical with four seasons (winter, summer, autumn and spring). Average monthly temperatures range from 25.1 °C in June to 18.1 °C in December. Average annual humidity is about 76,6%, with the highest (84%) in March and the lowest (70%) in December. During the year, total sunshine time is about 1075.2 h [17]. Our study is made of 2 distinct parts: First, a survey on production performances and meat quality. Production measures have been recorded continuously for 28 weeks and the meat quality has been assessed at slaughter. Second, the quantitative and qualitative egg production performances have been continuously monitored in laying phase during 52 weeks.

### 2.1. Growth, Carcass and Meat Quality

The experiment lasted from January 2017 to July 2017. A total of 250 chicks originating from the chicken flocks of Ho (62 males and 63 females) and Dong Tao (61 males and 64 females) have been used to survey the growth performance and meat production potential. These breeds are pure breeds and the animals come from the national programme of chicken conservation. We first randomly divided all the chicks into pens containing 25 individuals of the same breed (2 rows of 5 pens). The chicks were then floor-bred on a rice husks litter in the same ventilated building, with small brick walls and nets separating the pens. The chicks under 5 weeks of age were put under a heating lamp and the room temperature was regulated manually according to the chicks’ behaviour. The chickens after 5 weeks of age were raised under room condition (i.e., no control of the light, temperature and ventilation). The birds stayed in the same pens until slaughter and the males and females were not kept separate. All the chicks used the same feed (commercial animal feed pellets—Table 1), and we used the same vaccine protocol for all of them (Table 2). The feed was provided for chickens twice a day (6-7 am and 1-2 pm) using two 60 cm X 15 cm troughs per pen. Water was available in similar troughs (1/pen). Feed and water were always available. Mortality was low, with only 3 birds (two Dong Tao and one Ho) dying in the first week.

Each bird was identified individually, first using a numbered plastic ring at the leg, then using metallic ones after 5 weeks. At 5 weeks of age, sex was determined by the differences in comb size and wing feathers colour, which are sex-specific [15,16]. We weighed the chickens every week on a fixed day, from one to six weeks of age with an electronic scale (accuracy 0.01g). From 7 to 28 weeks of age, we used a mechanical scale (accuracy 5 g). The differences between final and initial weights were used to compute individual average daily gains (ADG) between the first week and 28 weeks. Feed intake was also recorded for each batch. The feed conversion ratio (FCR) was defined as the ratio of the average (for the batch) amount of feed ingested throughout the rearing period to the average body weight gained during that period. One cock and one hen were randomly sampled in each pen to be slaughtered at 12, 16, 20, 24 and 28 weeks of age (leading to slaughter n = 50 individuals in each breed). Feed was withdrawn approximately 12 h before slaughter. Slaughtered animals were bled out using the normal procedure in use in Vietnam (i.e., with a knife), plucked under warm water, weighed again and eviscerated. The legs were sectioned at the tibiotarsus-metatarsus joint and the head was cut at the skull atlas joint. The warm carcass was then weighed. The dressing out percentage was calculated as the ratio between warm carcass weight and live weight at slaughter. At this stage, the carcass was cut and the wings, legs and drumsticks were sampled and weighed after being skinned. The pectoral muscles (*Pectoralis major* and *Pectoralis profondus*) and the thigh muscle were sampled just after slaughter, weighed and packed in plastic bags for conservation, at 4 °C, for 24 h. Water loss was calculated as the difference between muscles weight at sampling and after 24 h of draining on absorbent paper. The pH was measured using a portable pH-meter (Testo 230 with an electrode type 03 pH, Germany). Three measurements were performed and the average of these readings was considered as the final pH value. The samples after 24 h of storage were cooked to internal temperature of 75 °C for 60 minutes. The post-processing losses were determined based on the difference in the weights of the sample before, and after, processing.

Meat toughness was determined on meat samples after processing. The standardized shear force perpendicular to the axis of muscle fibres was measured in Newton (N) using a Warner Bratzler machine 2000D (USA).

Finally, meat composition was assessed in terms of protein and fat content, dry matter and mineral content according to standard methods.

### 2.2. Egg Production and Quality

Laying performance was investigated using 36 Ho and 32 Dong Tao hens from June 1, 2016 to June 15, 2017. These hens came from farms involved in the national program of chicken conservation. They were reared on a (unknown) farm diet before being collected for the experiment at 22 weeks (Dong Tao) or 24 weeks (Ho) and caged directly upon arrival at VNUA. Ho and Dong Tao in this experimentation were purebred animals. All hens were raised in the same building and put in individual cages (cages dimensions: 40cm × 65cm × 38cm) about 15–20 days before laying the first egg. This system made it possible to obtain individual egg production and feed consumptions. Drinking water was provided automatically using nipples waterers and we used the same vaccination protocol for all hens (see Table 2). 140 g of feed mix (see Table 3) were provided daily to each individual hen in 2 meals (6–7 am and 1–2 am). The composition of feed for hens was analyzed in the Central laboratory of the Faculty of Animal Science, Vietnam National University of Agriculture. The crude protein, fat content, fiber, calcium and phosphorus were determined according to standard methods (TCVN 4328-2001; TCVN 4331-2001, TCVN 9590:2013, TCVN 6198-1996, and TCVN 1525-2001, respectively).

The amount of leftovers was weighed at the end of the day. The survey was carried out during 52 weeks of laying. All hens were inseminated artificially every two days using 0.05 mLsemen doses collected from 6 males from each breed. These males were randomly selected at 26 weeks for Dong Tao and 28 weeks for Ho from farms included in the conservation program. After 2 weeks training, semen was collected using massage into 1.5 ml Eppendorf tubes and hens were directly inseminated using doses of 0.05 mL (6 randomly chosen hens/cock). This process (sperm collection and insemination of all hens) was repeated every two days in order to maximize the fertilization percentage. Eggs were collected each day and stored at room temperature (ranging between 17.02 °C and 18.20 °C) for a maximum of 3 days. Abnormal, cracked or unshelled eggs were removed. Eggs analysis (see description below) took place every Wednesday with all eggs collected from Monday to Wednesday. All eggs, collected during the other days of the week, were incubated using a MacTech MT500 incubator.

Egg quality analysis was implemented on the collected eggs. In total, 1673 eggs of Ho and 1723 eggs of Dong Tao were collected, and the weekly total egg weights were obtained using an electronic scale (accuracy 0.01 g), while average egg weights were computed by dividing the total weekly eggs weights by the corresponding eggs number. Using a random subset of 108 eggs from each breed, we measured the length and width of the eggs with an electronic slider with a precision of 0.01 mm. The eggshell index was calculated as the ratio of the width to the length multiplied by 100 [18]. In order to minimize errors in the subsequent results, the re-examination of the eggs was carried out to ensure that the cracked eggs were definitely removed. We assessed the egg quality as described in [18]. Briefly, we computed the yolk/albumen ratio, and we measured the height of albumen and the height of yolk to infer the Haugh unit. The maximal breaking force (Fmax) of the eggshell was determined using the static compression method with a Universal tensile and compression test machine described in [18]. The shell thickness was measured at three different random points in the equatorial shell zone using an electronic micrometre (precision 0.01mm). The calculated average was used as the trait value. According to [18], the eggshell thickness is slightly thinner but more constant in the equatorial shell zone compared to other shell zones.

The eggs were then shredded carefully to remove the albumen, and the shell (including the membrane) and the yolk was weighed using an electronic scale (precision of 0.01g). The egg albumen weight was determined by taking the egg weight minus the shell and yolk weights.

The remaining eggs were also checked to remove abnormal eggs before incubation (eliminating cracked, broken, deformed eggs). All eggs were hatched using an automatic incubator with a capacity of 500 eggs per session. The incubator allowed temperature, humidity and automatic eggs rotation adjustments. The embryonated eggs were screened after 7 days of incubation and examined using a lamp light focused on the egg. The eggs with embryos, the number of chicks hatched, the deformed chicks hatched, the ratio of eggs containing embryos, and the ratio of chicks hatched were recorded.

### 2.3. Statistical Analysis

All statistical analyses were performed using the following linear mixed model, with slight adaptations when needed, as indicated below:

y_ijkl_ = µ + A_i_ + B_j_ + C_k_ + (AB)_ij_ + (AC)_ik_ + (BC)_jk_ + (ABC)_ijk_+ l_m_+ e_ijkl_

where:−y_ijkl_ represents the modelled trait (see below) measured on animal l with age i from breed j and sex k,−µ stands for the overall mean,−A_i_ is the fixed effect of age (i: 1, 4, 8, 12, 16, 20, 24 and 28 weeks),−B_j_ is the fixed effect of breed (j: Ho, Dong Tao),−C_k_ is the fixed effect of sex (k: male, female),−(AB)_ij_, (AC)_ik_, (BC)_jk_ represent two-ways interactions between age i and breed j, between age i and sex k, and between breed j and sex k, respectively,−(ABC)_ijk_ represents the three-ways interaction between age i, breed j and sex k,−l_m_ is the random effect of the pen m,−e_ijkl_ is a random residual effect for animal l with age i from breed j and sex k.

We analysed the results from the experiment on growth and meat quality as follows.

For the growth performances (weight and average daily gain ADG), we used the above model on individual measurements (25 individuals/pen, 10 pens). To account for the dependence between successive measurements on the same animal in this longitudinal study (up to 8 measurements/individual), we modelled the correlation between successive measurements on the same animal using a type 1 -autoregressive structure (using the Mixed procedure of SAS, version 9.3). Note that we nevertheless included a random pen effect to correct for the potential differences between pens.

We used the 10 pen averages of feed intake and growth to compute the weekly FCR. We then used the abovementioned mixed model after excluding the fixed effect of sex and its interactions, since observations were made on approximately sex-balanced batches of individuals.

For the carcass and meat quality traits, a fixed linear model was fitted (procedure GLM, SAS version 9.3) on individual measurements (one male and one female from each pen at each of five time points) assuming homoscedastic and uncorrelated normally distributed residuals.

The results from the experiment on egg production and egg quality were analysed using linear models including the fixed effect of breed only. For the laying performances, we considered the traits measured on 36 Dong Tao and 36 Ho randomly chosen hens. For the hatching performances, we analysed the results from incubator runs (1 measure/run/trait). An incubator run involves all the eggs laid in the 3 preceding days. Finally, the egg quality was investigated using the measures made on 108 randomly sampled eggs from each breed.

For all analyses, least squares means (LSM), standard error (SE), and tests of the differences between levels of an effect were obtained, with significance level set at *p* < 0.05 after correcting for multiple testing using Bonferroni procedure. Averages reported in the Results section are LSM.

## 3. Results

### 3.1. Growth, Carcass and Meat Quality

#### 3.1.1. Growth

Sex and age had effects on the growth (*p* < 0.0001), with heavier males than females and weights increasing with age. The interaction between sex and age demonstrated a clear difference in the growth in both sexes (*p* < 0.001): Starting from similar weight up to week 4 (*p* > 0.05), the weights start to diverge and become different (*p* < 0.0001) from week 8 onwards. We did not observe any main breed effect (no global differences between Ho and Dong Tao individual weights, *p* > 0.05), nor any differences between the weights in the two breeds with time (comparison at the various time points all provided non-significant differences, *p* > 0.05). Within each sex, the growths were also similar (the sex*age*breed interaction was not significant (*p* = 0.7769), and comparisons of males (or females) from the 2 breeds at different ages did not produce any significant differences). Nevertheless, we found a slight (*p* = 0.0182) interaction between breed and sex: the weight of Ho males was slightly higher than the weight of Dong Tao males, while the opposite was true for females (although these two differences are not significant). Figure 3 summarizes these results.

For ADG, the random pen effect variance was close to 0, and this effect was therefore removed from the model. Figure 4 illustrates the results from this analysis. 

We found effects of the sex on ADG, males grew quicker than females (*p* < 0.0001). ADG increased between weeks 4 and 16 (*p* < 0.0001), stabilized during weeks 16 to 20 (*p* = 0.0728), then started to decrease between weeks 20 and 24 (*p* = 0.0139) and decreased more rapidly between weeks 24 and 28 (*p* < 0.0001). We found also an interaction between these two effects (sex and week) (*p* < 0.0001). As was the case for weights, ADG were similar for males and females during the first 4 weeks (*p* = 0.7585), but then started to differ from week 8 to week 28 (all p-values < 0.001). No breed difference was visible (*p* = 0.4513).

We used the week averages to calculate FCR at the various time points. FCR was increasing (*p* < 0.0001) with the age of the animals, with slight differences between breeds over time (*p* = 0.0133), but no global difference between breeds (*p* = 0.9821) as shown on Figure 5. More precisely, FCR for Dong Tao was higher than for Ho at week 20 (*p* = 0.0179), while the reverse was true and more marked at week 28 (*p* = 0.0038).

#### 3.1.2. Carcass Yield and Meat Quality

The results of the statistical analyses on the carcass characteristics and meat quality of Ho and Dong Tao chickens at different ages are presented in Table 4. The means and standard errors for these traits are shown on Appendix A.

Looking first at weights and yields (Appendix A), it turned out that all the weight traits (body weight, carcass weight, thighs meat weight, pectoral meat weight) increased (*p* < 0.0001) with the age of the animal, while yield first increased (*p* < 0.0001) between weeks 12 and 16, then stabilized to finally decrease back to its initial value at week 28.

Sex was another important factor: Body weight, carcass weight, yield, thighs meat weight, shear force of pectoral muscle, shear force of thighs meat and pH 24h of thighs meat were higher for males than for females (*p* < 0.0001 except for pectoral muscle weight, where *p* < 0.01). Sex by age interactions were also significant (*p* < 0.01 for yield and pectoral muscle weight, *p* < 0.0001 for the other traits), indicating different profiles for males and females with respect to age. Breed was also important. Although, no difference existed between the body weights of Ho and Dong Tao animals (*p* > 0.05), Ho had a higher yield than Dong Tao (*p* < 0.0001), higher carcass and thighs weight (*p* < 0.0001) but lower pectoral weight (*p* < 0.05). The profiles of all the traits except pectoral weight also differed between the breeds (*p* < 0.05 for yield, *p* < 0.01 for the other traits).

Next, in considering meat quality, we found an increase in the shear forces for pectoral and thighs meat with the age of the animals (*p* < 0.0001) (Table 4; Appendix A). These forces were also larger for Dong Tao than for Ho (*p* < 0.01 for pectoral muscle, *p* < 0.05 for thighs muscle), and for females than for males (*p* < 0.01 for pectoral muscle, *p* < 0.0001 for thighs muscle). An interaction between sex and breed informed that these results must be considered with caution. The difference between males and females for the shear force in thighs is due to the Ho individuals, the Dong Tao chickens showing no such difference (*p* = 0.1337).

Drip loss is another aspect of meat quality. No main effect of breed or sex was found for this trait, either in pectoral or in thighs muscles, and after 24 h or after cooking (*p* < 0.05). Nevertheless, behaviour was different between Ho and Dong Tao for pectoral muscle drip loss: while values in Dong Tao increased when going from males to females, values decreased in Ho, after 24 h and after cooking. Age also affected the drip loss, with losses increasing with the age of animals older than 16 weeks for pectoral meat after 24 h (*p* < 0.01), thighs meat after 24 h (*p* < 0.001) and after cooking (*p* < 0.0001). For this last trait, the situation was more difficult. While, the values remained stable in Dong Tao, the value increased with the age in Ho, generating an interaction (*p* < 0.0001) between these two factors. Furthermore, this interaction differed between males and females, leading to a three-ways interaction (*p* = 0.0006).

Next, the age of the animals also turned out to affect the ultimate pH (i.e., after 24 h), with values decreasing when the age of the animals increased (*p* < 0.0001 for pectoral and thighs meat). The pH was higher in Ho than in Dong Tao for pectoral meat (*p* = 0.0029), but not for thighs meat (*p* = 0.327) (Table 4; Appendix A). Although, there was only a mild effect of sex on pH in thighs meat (*p* < 0.05), the main difference was due to the Ho chickens, where the difference between males and females was strong (*p* < 0.0001), while not present in Dong Tao (*p* > 0.05). At 28 weeks, the values for the Ho and the Dong Tao breeds were found close to the research results of [3,19]. The pH after 24 h of H’mong breed was similarly recorded as 5.8 for pectoral muscles and between 5.8 for females and 5.9 for males for thighs meat muscles [20]. The values between 5.57 and 5.76 have been recorded for the meat pH of a Korean indigenous chicken breed [21]. According to [19], the pH after 24h of Dong Tao was 5.42 for females and 5.78 for males for the pectoral muscle, and 5.67 for males and 6.04 for females for the thighs muscles. For the pH of commercial broilers breeds (Ross 308, Cobb 500 and Cobb 800), respective values of 5.74, 5.65 and 5.67 are reported in [22].

The meat content of the pectoral muscle differed (*p* < 0.0001) between the breeds, with higher values for dry matter, fat and protein contents for Ho, but larger values for mineral content in Dong Tao. Globally, the dry matter content increased between weeks 12 and 16 (*p* < 0.01), and then remained relatively stable, while we observed a more continuous increasing trend in fat and protein content (*p* < 0.0001), only stabilizing after 24 to 28 weeks. No such trend was visible for the mineral content (*p* > 0.05). Mineral and fat contents were higher in female meat than in males (*p* < 0.0001), while differences in protein (*p* > 0.05) and dry matter (male values higher than females, *p* = 0.0490) were barely or not significant. As reported in Table 5, a few interactions between breed, sex and age effects made the picture a little bit more complicated. For example, looking at protein content in Table 5 showed that, although the main effects reported above were present, the situation was a bit more complex, with for example different trends with the age in Ho and Dong Tao. The age at slaughter was another important factor for the meat protein content (*p* < 0.0001) and fat content (*p* < 0.0001), with values globally increasing with the age. For the protein content and the dry matter, the increasing trend with the age at slaughter was slightly different for the two breeds, with values for Ho increasing slightly more than values for Dong Tao. At week 28 (normal slaughter age in these breeds), the protein content in Ho and Dong Tao were higher than the protein content found in H’mong breed (19.7% for thighs meat) (20.4% for pectoral meat) [20]. It was also higher than the protein contents found in popular broiler lines meat, where Ross 308, Cobb 500 and Cobb 800 showed values of 21.9%, 22.4%, and 22.8%, respectively [22]. Sex also affected the meat composition. At week 28, the fat content percentage in Ho and Dong Tao males meat did not differ (*p* = 0.10), but differed (*p* < 0.05) from the females values. It was also lower than the values observed in the H’mong breed, where values of 0.4% in the pectoral meat and of 1.4% in thighs meat have been obtained [20]. The breed and the age at slaughter of the animals also affected dry matter. Average values in the Ho breed were higher than in the Dong Tao breed (*p* < 0.0001). Furthermore, the dry matter increased between 12 weeks (25.41% in Ho, 25.42% in Dong Tao) and 16 weeks, almost stabilizing in older animals, with values between 25.63% and 26.87% in Dong Tao, and 26.89% and 27.41% in Ho, illustrating a slightly larger increase in Ho (*p* = 0.01).

### 3.2. Egg Production and Egg Quality

#### 3.2.1. Egg Production

Table 6 and Table 7 report the laying performances comparison for the two breeds. In these analyses, we modelled only the breed in the statistical analyses.

Table 6 presents the results for a random sample of 36 Ho and 36 Dong Tao hens. For the feed traits, the sample number was 36 for Ho and 32 for Dong Tao hens. The laying age of the first egg for Ho hens was close to one month later than that of Dong Tao hens (*p* < 0.0001), and the body weight at the first egg for Ho hens was higher than for Dong Tao hens (*p* < 0.01). Nevertheless, the laying performance measured as the number of eggs laid within 52 weeks was similar for the two breeds (*p* > 0.05). 

The feed offered per day and the total feed offered over 52 weeks was similar for both breeds (*p* > 0.05). The feed needed to produce 10 eggs was also similar for the two breeds (*p* > 0.05), and so was the amount of feed needed to produce 1g of egg, the feed conversion rates being close to 9 grams per gram in both breeds.

Table 7 shows the results for batches of incubated eggs, each batch corresponding to a run of the incubator. Since the average number of incubated eggs was higher for Dong Tao than for Ho (*p* < 0.001), only rates are meaningful. The ratio of embryonated eggs was much higher (*p* < 0.0001) in Dong Tao than in Ho, and the mortality (% dead embryonic eggs) was also higher in Dong Tao, either when comparing the number of eggs (*p* < 0.001) or the percentages of dead among the incubated eggs (*p* < 0.01). However, comparing the dead percentages among the embryonated eggs did not reveal any difference (*p* = 0.1380). Similarly, for hatching, although the number of hatched chicks (*p* = 0.0002) and the percentage of hatched chicks among incubated eggs (*p* = 0.0017) were higher in Dong Tao than in Ho, the percentages of hatched chicks among the embryonated eggs were similar (*p* = 0.6060). Finally, the percentage of malformed chicks was higher in Ho than in Dong Tao, but the difference was not significant (*p* = 0.1704), which might either indicate an absence of difference or a lack of statistical power for that trait.

#### 3.2.2. Egg Quality

Table 8 reports the mean values for the egg quality traits for both breeds.

Although, the egg and albumen weights were similar in both breeds (*p* = 0.3230, and *p* = 0.1750, respectively), the yolk weight was higher in Dong Tao eggs (*p* < 0.0001) and the eggshell weight was lower (*p* = 0.0007). Consequently, the proportions of these components varied across breeds (*p* < 0.0001 for all three percentages) and the yolk to albumen ratio was higher in Dong Tao (*p* < 0.0001). Haugh units, which measure the protein content and the freshness of the eggs, were not different between the two breeds (*p* = 0.1311).

Looking at the eggshells, the thicknesses were found similar in both breeds (*p* = 0.2035), but the shapes differed, as demonstrated using the eggshell shape index: Ho eggs are more spherical than Dong Tao ones (*p* < 0.0001). Additionally, and maybe consequently, the force needed to break the eggshell was higher for Ho than for Dong Tao eggs (*p* = 0.0032).

## 4. Discussion

Our growth results in Ho are comparable to those from previous results on the same breed [13,19]. We have demonstrated once again that Ho and Dong Tao are indigenous chicken breeds with large sizes, compared to other indigenous chicken breeds in Vietnam, such as H’mong or many toes chickens [20,23]. This also demonstrates that the growth performance of chickens can be influenced by many factors including feed and sex, but also breed [21]. 

Large body weight is one of the least advantageous factors in Vietnam’s commercial’s market because Vietnamese consumers prefer small size chickens [9]. This raises questions with respect to the presence of these large size breeds. As mentioned earlier, the Ho and the Dong Tao were historically the two breeds used as gifts for the King [11]. Most likely, their large body weight was one of the important criteria for their selection as a royal gift [8]. A traditional festival used to take place each year to select the most beautiful individual from the Ho breed and the owner of the best bird was always awarded a prize. This festival is still taking place every year. According to morpho-biometric criteria, the body weight index is very important in that respect [11,15].

Considering the entire testing period, the FCR of Ho and Dong Tao broilers were very similar, with only few minor differences. Comparing these results to similar results from other breeds shows how inefficient Ho and Dong Tao are, even when compared to indigenous breeds [20,23]. As can be expected, the native breeds have slower growth performance and poorer feed conversion ratio than hybrid broilers [24,25].

We observed different patterns with time for the carcass weights and yield percentages of Ho and Dong Tao animals. We can find a possible origin of this difference by looking at the carcass composition in more details. Globally, the thigh meat weight is larger for the Ho animals than for the Dong Tao ones, with this difference increasing with the age of the animals. In contrast, pectoral meat weight tends to be higher for middle-aged Dong Tao animals, but this difference vanishes in the later stages. These differential muscular growths might explain the patterns observed for the carcass weights and yields. We should, nevertheless, emphasize that these patterns of muscle weight evolution with the age of the animals are sex-dependent within each breed, but this interaction between sex and age does not differ between the two examined breeds.

The thighs meat weight and pectoral muscle weight for Ho at week 12 reported in our study are lower than the results of an already published study on the same breed by [13]. Our results, along with many other results on indigenous breeds (e.g., [23]) show again that many factors, including the breed, the sex, but also environmental factors, including feed, and the slaughtering conditions [21,26] influence carcass characteristics. Vietnamese consumers prefer Ho and Dong Tao meat from animals slaughtered at the age of 28 weeks. People estimate that the meat at this age has a more consistent taste, which might be due to the flavour and taste improving with the age of the animals [3]. At this age of 28 weeks, the tenderness of the Ho and Dong Tao meat has, nevertheless, decreased, since the shear forces for the pectoral and the thighs muscle consistently increase with the age at slaughter, as we show in our study, as well as in other similar works [19]. In comparison, for commercial broilers, the shear force of pectoral muscle at the time of slaughter for several hybrid lines ranged between 10.9 N and 12 N [22]. Optimal slaughtering time is between six and nine weeks for these breeds, so we cannot provide an exact comparison. However, the earliest slaughtering time in our experiment (12 weeks) led to average values of 23.16N and 23.64N for Ho, and Dong Tao, respectively, which is definitely much higher than these commercial standards. Genetic differences between breeds and the slaughter age can affect the tenderness of chicken meat [27,28]. Our study supports these hypotheses, showing differences between breeds, and a decrease of the tenderness with the age at slaughter.

Water retention capacity is another important feature of meat quality. If the water retention capacity of meat is low, then meat and meat products lack sweetness [21,29]. We have observed that drip loss is strongly influenced by the breed, the age at slaughter and the sex of the animal, according to a complex interactions pattern. The exact mechanisms underlying these complex behaviours could be investigated. For example, questions including why the drip loss remains stable with the slaughter age in Dong Tao, while it sharply increases in Ho (especially in females), might be asked and might provide clues to water retention mechanisms in the meat. The analysis of the composition of the pectoral muscles of Ho and Dong Tao animals showed that the protein content, fat content and dry matter of Ho animal muscles were globally higher than those of Dong Tao animals. This different composition might be an explanation for the different properties of the meat observed above.

In our experimental conditions, the egg production was similar for the two breeds. Although relatively low, with an average close to one egg every four days, the production of Ho was higher than in an earlier study where the egg production recorded on the field was 66.18 eggs/year [15]. Similarly, the egg production of Dong Tao was also higher than previously reported results (55 to 66 eggs/year) [10]. Among the indigenous Vietnamese breeds, the egg production of Ho and Dong Tao was lower than for some breeds such as Ri (123 eggs) and Tau Vang (120 eggs), but higher than for others like the Mia breed (55–60 eggs). Differences in raising, feed and genetics are possible explanations for these differences, pointing to avenues for improvements. Anyway, the reproductive performances of the Ho and Dong Tao breeds are low, and their eggs are not used as a commercial food. Eggs are often used for hatching to replace the flock [9,10].

The efficiencies, measured by the amount of feed, needed to produce 10 eggs or, alternatively, by the amount of feed needed to produce one gram of egg, were similar for the two breeds, and expectedly rather low for these low producing animals, with values in the range of 9 grams of feed for one gram of egg. Although, these breed are not efficient at converting feed into eggs, they do better than, for example, the H’mong breed, for which 6580 g of feed were needed to produce ten eggs with an average weight of 38.10 g, corresponding to more than 17 g of feed for 1g of egg [30].

Since eggs are mostly used for hatching, the percentages of embryonated and hatched eggs are important traits for these breeds. In that respect, Dong Tao perform better than Ho chickens, with a higher percentage of hatching eggs. The main origin of this difference lies in the rate of embryonated eggs for the two breeds, where Dong Tao succeed to get embryos in 78.60% of the eggs, while Ho only do so for 62.00%. After getting embryos, hatching rates are similar for the 2 breeds. The reason for this differential success should be investigated, and traits like cock sperm quality might deserve attention to try to solve this problem. Furthermore, although the percentage of malformed chicks for both breeds are not different, the rate of malformations is higher in Ho, which exacerbates the problem.

The egg weights of Ho and Dong Tao are similar, and larger than most of the weights met in the other indigenous breeds [9,20,23]. To compare to commercial layer lines, weights range from 68.3 to 69 grams for the popular Brown breed [31]. This large difference with indigenous (and barely selected) lines come from the intense directional selection acting on commercial hybrid laying hens [32]. This indicates that potential improvements of the indigenous breeds performance could come from a more organized selection [33,34], after showing that the weight and the eggs composition have genetic components.

A high yolk/albumen ratio is an important criteria in the egg production industry, as it can improve the egg quality for consumers and for the food processing industry [32]. Although the egg weight of hybrid laying hens is usually higher than that of the indigenous breeds, the yolk/albumen ratio is generally lower or equal [3]. In our study, the yolk weight, the yolk percentage and the yolk/albumen ratio of Dong Tao are significantly higher than those of Ho. Interestingly, the yolk weight of Ho and Dong Tao are also higher than for Ri [9] and the yolk to albumen ratio is higher than in indigenous Korean chickens breeds [21]. In summary, Ho and Dong Tao produce big eggs with interesting composition when compared to other indigenous breeds.

Eggshell strength is another important factor in the egg production industry [3,35] Fragility of the eggshell is responsible for about 6-8% loss in the egg industry [36]. The average force needed to break a Ho’s egg was significantly higher than for a Dong Tao’s egg, but also significantly lower than for a Ri’s egg [9]. Values ranging from 30.9 N to 37.8 N have been reported for six commercial lines, not very different from the values observed in our study [37]. This shows that the eggs from indigenous breeds do not seem more fragile than those commonly produced in industrial production units.

Eggshell thicknesses for Ho and Dong Tao were similar, and smaller than Ri [9] and H’mong [30]. Eggshell thickness is closely, and logically, related to the maximal force needed to break an egg [35], and is therefore, an important parameter. In our study, lower maximal forces needed to break Ho and Dong Tao eggs indeed corresponded to thinner eggshells in these two breeds when compared to Ri’s. The addition of calcium may improve eggshell thickness without affecting eggshell ratio [31].

Finally, our measures of Haugh units are useful to describe egg protein quality and freshness [32]. According to USDA, best quality eggs (AA eggs) should have Haugh units above 72. The freshness of indigenous breeds eggs are commonly above this threshold. In our study, Ho and Dong Tao eggs produced similar values, higher than the observed values for Ri (76.14–77.67) and lower than the HU index value for H’mong (89.42) [30]. In comparison, values for Barred Plymouth Rock, White Leghorn, Rhode Island Red and White Rock layers in Bangladesh ranged between 45.81 and 58.68, considerably lower than for the indigenous breeds described above [38].

## 5. Conclusions

This study has attempted to show the strengths and weaknesses of two emblematic chicken breeds from North Vietnam. As commonly the case with indigenous breeds, production performances of Ho and Dong Tao animals were poor, when compared to current industrial standards. One of the main reasons for this low productivity is the absence of an organized selection program, partly because of the small population size, but maybe also because selection objectives differ from the specialized lines met in commercial companies today. For example, adaptation to the indigenous breeding conditions is probably a more important objective than quantitative production. On the other hand, this quasi-absence of selection has also advantages. As commonly known, strong directional selection may strongly affect the genetic diversity, which in turn, might be harmful for animal resilience with respect to environmental changes or if new selection objectives are pursued. Note nevertheless that the populations targeted in this study are small, with consequently drift potentially acting adversely on the genetic diversity. Therefore, a genomic study is needed to bring more information on the current genetic status of these breeds.

Reproduction performances were also weak. A comparison of the field results to the performances recorded in our controlled experience shows that there is ample room for improvement. Providing farmers training programs in breeding, husbandry practices, housing, nutrition, record keeping etc., could certainly be helpful. Another possible way to improve the situation would be to provide improved animals to the breeders, given that changing breeding practices might be a difficult task in the current context. Genetic selection and distribution of the selection products to interested farmers might help achieving this goal. Although, some reproduction traits are usually weakly heritable. In view of the low proportion of embryonated eggs, especially in Ho, preliminary work on traits such as sperm quality should be conducted first. Such studies would be welcome to contribute to the program of conservation and sustainable exploitation of these two breeds.

## Figures and Tables

**Figure 1 animals-10-00408-f001:**
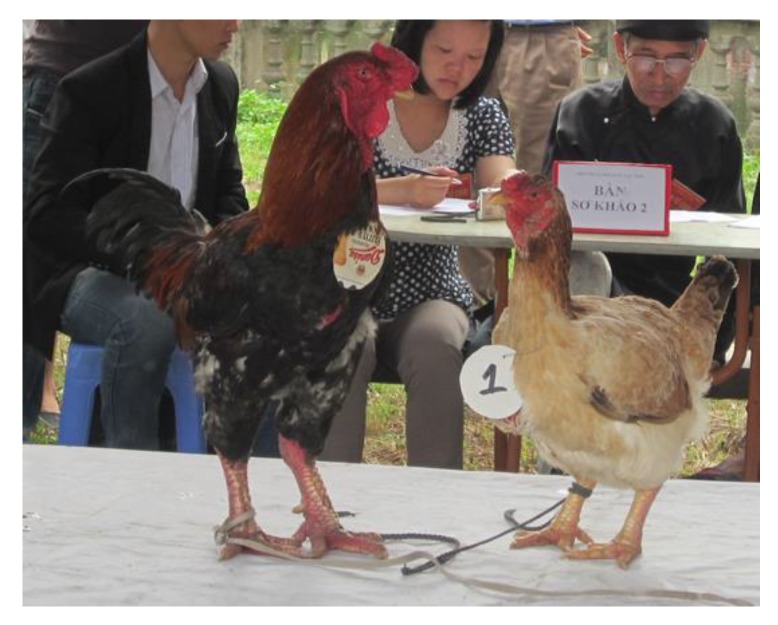
Adult female (**right**) and male (**left**) Ho chickens.

**Figure 2 animals-10-00408-f002:**
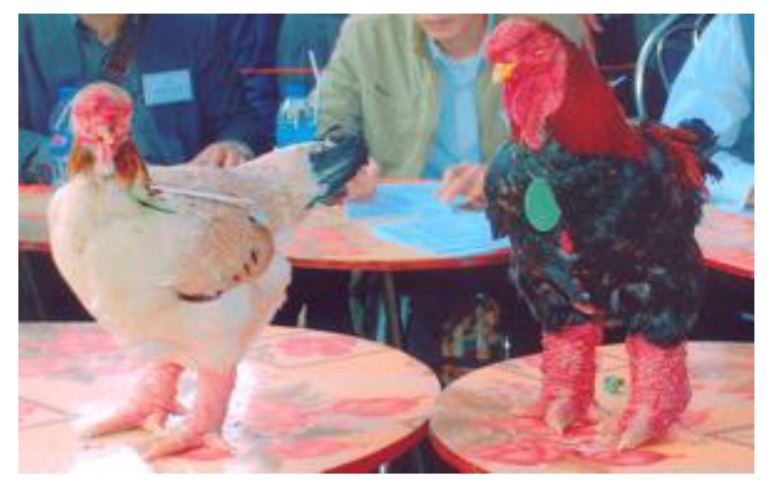
Adult female (**left**) and male (**right**) Dong Tao chickens.

**Figure 3 animals-10-00408-f003:**
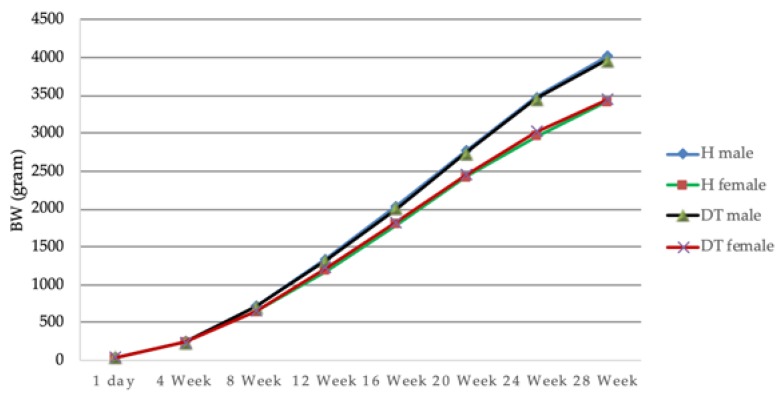
Growth performance of Ho and Dong Tao chickens, H: Ho breed, DT: Dong Tao breed.

**Figure 4 animals-10-00408-f004:**
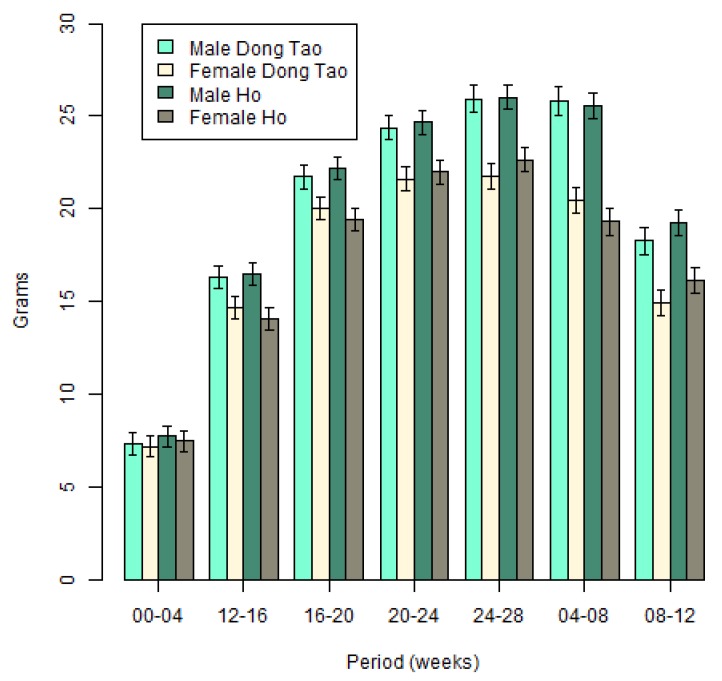
The average daily gains of Ho and Dong Tao chickens.

**Figure 5 animals-10-00408-f005:**
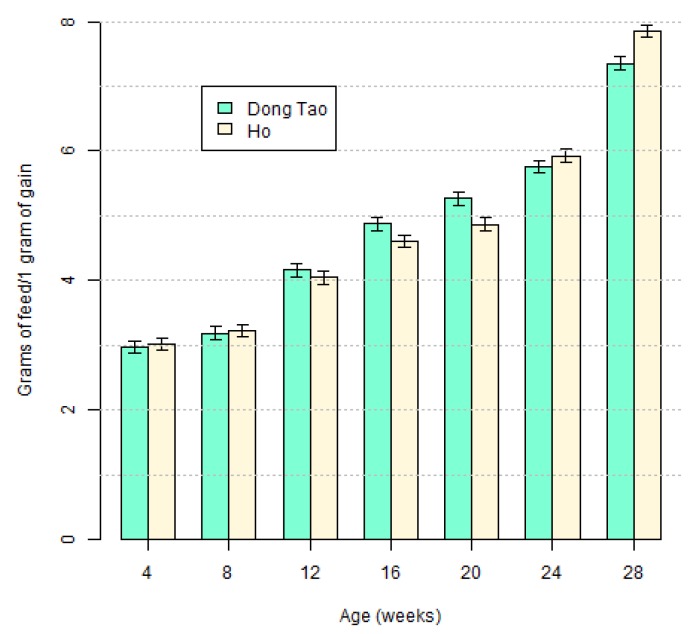
The feed conversion ratio (FCR) of Ho and Dong Tao chickens.

**Table 1 animals-10-00408-t001:** Diet composition for broilers from the first day of age to slaughter.

Ingredients (%)	Proportion			
	1–14 Days	15–28 Days	29–42 Days	43 Days-slaughter
Corn	59.80	62.90	65.40	68.50
Soybean meal	26.82	23.87	25.22	18.82
DDGS	3.00	2.00	0.34	1.00
Wheat DDGS	3.00	3.00	-	1.00
Limestone powder	1.50	1.40	1.40	1.20
Basa fish oil	1.20	1.20	1.98	1.22
Poultry vitamix (CM1976)	0.10	0.10	0.10	0.10
Canola	1.00	-	-	-
Dicalcium phosphate (DCP 17%)	0.98	0.73	0.53	0.12
Lysine 70%	0.70	0.60	0.45	0.46
Meat and bone meal	0.55	0.57	1.00	2.00
Premix	0.49	0.88	0.87	0.86
Salt powder (Nacl)	0.34	0.29	0.26	0.21
L-met pro (Methionine 90%)	0.22	0.18	0.21	0.20
Poultry minemix (CM4089)	0.15	0.13	0.13	0.13
Sodium bicarbonate 99%	0.05	0.05	0.05	0.10
Choline chloride 60%	0.05	0.05	0.05	0.05
Salinomycin 12%	0.05	0.05		
Rapeseed meal	-	2.00	2.00	4.00
Threonine. L 98%	-	-	0.01	0.03
**Analytical composition**				
Crude protein (%)	22.5	21.5	20.5	19
Calcium	1.5	1.5	1.15	0.8
Phosphorus	1	1	0.7	0.5
Crude fiber	5	5	5	5
Metabolizable energy (kcal/kg)	3000	3000	2950	3030

**Table 2 animals-10-00408-t002:** Vaccination protocol for Ho and Dong Tao breed.

Age (day)	Vaccine	Way
1	Marek’s disease	Subcutaneous injection
3	Newcastle	Oral vaccination
8	Gumboro	Subcutaneous injection
15	Newcastle	Oral vaccination
60	Newcastle	Subcutaneous injection
90	Fowl cholera	Subcutaneous injection
150	Newcastle	Subcutaneous injection

**Table 3 animals-10-00408-t003:** Diet composition for laying hens.

Ingredients (%)	Proportion
Corn	34.00
Paddy	33.00
Commercial animal feed concentrate ^1^	16.50
Rice bran	16.00
Minerals–vitamins ^2^	0.50
**Composition**	**Quantity**
Crude protein (g/kg)	142.50
Fat content (g/kg)	53.46
Crude fiber (g/kg)	42.21
Calcium (g/kg)	11.52
Phosphorus (g/kg)	5.90
Metabolizable energy (kcal/kg)	3030.12

^1^ Protein-rich foods, crude protein 45%; ^2^ Vitamin A: 3,000,000 UI/kg, vitamin D3: 30,000 UI/kg, Vitamin E: 100 mg/kg. The corn and the rice bran were flour. The paddy (rice) is primitive. All ingredients are mixed together using a mixer similar to a small cement mixer before distribution to the hens.

**Table 4 animals-10-00408-t004:** Effect of Breed (B), Sex (S) and Age (A) on carcass and meat quality traits (balanced design, with n = 5 observations per cell B*S*A (i.e., 1/pen)).

Variable	Significance Level						
	B	S	A	B*S	B*A	S*A	B*S*A
Body weight (g)	NS	***	***	NS	**	***	NS
Carcass weight (g)	***	***	***	NS	**	***	NS
Yield (%)	***	***	***	NS	*	**	NS
Thighs meat weight (g)	***	***	***	NS	**	***	NS
Pectoral muscle weight (g)	*	**	***	NS	NS	**	NS
Shear force of pectoral muscle (N)	**	**	***	NS	NS	NS	NS
Shear force of thighs meat (N)	*	***	***	***	NS	NS	NS
Drip loss of pectoral meat (%)	NS	NS	**	**	NS	NS	NS
Drip loss of thighs meat (%)	NS	NS	***	NS	NS	NS	NS
Cooking loss of pectoral (%)	NS	NS	NS	***	NS	NS	NS
Cooking loss of thighs meat (%)	NS	NS	***	NS	***	NS	***
pH 24h of pectoral meat	**	NS	***	NS	*	NS	NS
pH 24h of thighs meat	*	*	***	**	NS	NS	NS

Significance levels: *** *p* < 0.001; ** *p* < 0.01; * *p* < 0.05; NS: *p* ≥ 0.05.

**Table 5 animals-10-00408-t005:** Significance of the effects in the statistical model (upper part) and chemical composition of pectoral muscle of Ho and Dong Tao breed (B) by age (A), Sex (S) (LSM ± SE).(lower part).

p-value	B	S		A	B*S		B*A	S*A		B*S*A
**Prot (%)**	***	NS		***	NS		**	NS		*
**Fat (%)**	***	***		***	NS		NS	NS		NS
**Miner.**	***	***		NS	NS		**	*		*
**Dry mat**	***	*		***	NS		*	NS		NS
**Age (week)**	**Ho (n = 5)**					**Dong Tao (n = 5)**				
	**Male**		**Female**			**Male**			**Female**	
**Protein (%)**										
12	24.60 ± 0.33		23.81 ± 0.33			23.06 ± 0.33			23.68 ± 0.33	
16	24.52 ± 0.33		25.28 ± 0.33			24.12 ± 0.33			24.32 ± 0.33	
20	25.25 ± 0.33		25.20 ± 0.33			23.72 ± 0.33			22.14 ± 0.33	
24	25.55 ± 0.33		25.62 ± 0.33			24.92 ± 0.33			23.94 ± 0.33	
28	25.62 ± 0.33		25.32 ± 0.33			25.26 ± 0.33			24.74 ± 0.33	
**Fat (%)**										
12	0.51 ± 0.11		0.61 ± 0.11			0.17 ± 0.11			0.54 ± 0.11	
16	0.57 ± 0.11		1.07 ± 0.11			0.56 ± 0.11			0.82 ± 0.11	
20	0.70 ± 0.11		1.36 ± 0.11			0.64 ± 0.11			1.12 ± 0.11	
24	0.91 ± 0.11		1.56 ± 0.11			0.77 ± 0.11			1.16 ± 0.11	
28	1.07 ± 0.11		1.45 ± 0.11			0.81 ± 0.11			1.18 ± 0.11	
**Minerals**										
12	1.19 ± 0.06		1.48 ± 0.06			1.61 ± 0.06			1.56 ± 0.06	
16	1.15 ± 0.06		1.62 ± 0.06			1.47 ± 0.06			1.61 ± 0.06	
20	1.20 ± 0.06		1.26 ± 0.06			1.66 ± 0.06			1.67 ± 0.06	
24	1.24 ± 0.06		1.27 ± 0.06			1.68 ± 0.06			1.79 ± 0.06	
28	1.29 ± 0.06		1.32 ± 0.06			1.65 ± 0.06			1.81 ± 0.06	
**Dry matter of meat**										
12	25.58 ± 0.42		25.23 ± 0.42			25.52 ± 0.42			25.33 ± 0.42	
16	26.70 ± 0.42		27.19 ± 0.42			27.30 ± 0.42			26.45 ± 0.42	
20	27.19 ± 0.42		27.24 ± 0.42			25.77 ± 0.42			25.51 ± 0.42	
24	27.25 ± 0.42		26.53 ± 0.42			25.80 ± 0.42			25.46 ± 0.42	
28	28.01 ± 0.42		26.81 ± 0.42			26.31 ± 0.42			25.91 ± 0.42	

Significance levels: *** *p* < 0.001; ** *p* < 0.01; * *p* < 0.05; NS: *p* ≥ 0.05.

**Table 6 animals-10-00408-t006:** Laying performances of Ho and Dong Tao breed.

Variable	Ho		Dong Tao		*p*-Value
	n^1^	LSM ± SE	n^1^	LSM ± SE	
Age at the first egg (day)	36	196.47 ± 1.48	36	168.36 ± 1.48	***
Body weight at the first egg (g)	36	2891.67 ± 26.45	36	2761.11 ± 37.21	**
Number of eggs/hen/52 weeks	36	88.47 ± 4.14	36	94.92 ± 4.14	NS
Feed/hen/day (g)	36	106.42 ± 0.69	32^2^	105.79 ± 0.73	NS
Feed/hen/10 eggs (g)	36	4682.83 ± 280.00	32	4456.65 ± 296.98	NS
Feed conversion ratio (FCR) (g/g)	36	9.11 ± 0.54	32	8.62 ± 0.58	NS

^1^ number of laying hen; ^2^ 4 hens have been discarded for loss of follow-up; significance levels: *** *p* < 0.001; NS: *p* ≥ 0.05.

**Table 7 animals-10-00408-t007:** Hatching performances of Ho and Dong Tao breed (LSM ± SE).

Variable	Ho (n = 46)^1^	Dong Tao (n = 47)^1^	*p*-Value
Number of incubated eggs (NI)	27.15 ± 1.80	36.36 ± 1.78	***
Number of embryonated eggs (NE)	17.47 ± 1.56	28.55 ± 1.55	***
Ratio NE/NI (%)	61.99 ± 2.57	78.59 ± 2.54	***
Embryonic mortality eggs (ND)	1.76 ± 0.46	4.81 ± 0.46	***
Ratio ND/NI (%)	7.97 ± 1.74	14.61 ± 1.72	**
Ratio ND/NE (%)	13.49 ± 2.25	18.20 ± 2.20	NS
Number of chicks hatching (NH)	13.56 ± 1.55	21.94 ± 1.54	***
Hatchability NH/NI (%)	45.90 ± 2.86	58.95 ± 2.83	**
Hatchability NH/NE (%)	73.14 ± 2.74	75.13 ± 2.68	NS
Chicks malformation (%)	4.34 ± 1.29	1.84 ± 1.27	NS

^1^ number of incubator runs; significance levels: *** *p* < 0.001; ** *p* < 0.01; NS: *p* ≥ 0.05.

**Table 8 animals-10-00408-t008:** Egg quality of Ho and Dong Tao chicken (LSM ± SE).

Variable	Ho (n = 108)^1^	Dong Tao (n = 108)^1^	*p*-Value
Egg weight (g)	51.43 ± 0.18	51.69 ± 0.18	NS
Albumen weight (g)	28.13 ± 0.14	27.86 ± 0.14	NS
Yolk weight (g)	16.76 ± 0.08	17.44 ± 0.08	***
Eggshell weight (g)	6.55 ± 0.03	6.39 ± 0.03	***
Albumen (%)	54.68 ± 0.14	53.88 ± 0.14	***
Yolk (%)	32.59 ± 0.12	33.75 ± 0.12	***
Eggshell (%)	12.73 ± 0.06	12.37 ± 0.06	***
Yolk/albumen ratio	0.60 ± 0.004	0.63 ± 0.004	***
Eggshell thickness (mm)	0.23 ± 0.001	0.22 ± 0.001	NS
Eggshell shape index	79.34 ± 0.29	75.82^b^ ± 0.29	***
Force max (N)	35.28 ± 0.31	33.97 ± 0.31	**
Haugh Units (HU)	81.53 ± 0.29	82.15 ± 0.29	NS

^1^ number of specimens; significance levels: *** *p* < 0.001; ** *p* < 0.01; NS: *p* ≥ 0.05.

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
