# Peer review of "Productive Performance and Egg and Meat Quality of Two Indigenous Poultry Breeds in Vietnam, Ho and Dong Tao, Fed on Commercial Feed"

_animals, 2020, doi:10.3390/ani10030408_

Round 1
Reviewer 1 Report
This article has the aim of describing growth, egg production and meat and quality traits of two local Vietnamese chicken breeds: Ho and Dong Tao. The subject is interesting and somewhat important given that Vietnam is one of the chicken domestication centers. However, before it can be considered for publication it must be thoroughly revised. My indication is a major revision. Results and Discussion need to be completely revised.
The main issues are:
Sample size for egg production was small (N = 32 + 36 = 68 hens); Methods used for feed analysis should be presented; The results and discussion section on growth performance and carcass traits is long, confusing and at the same time incomplete. The way the results were presented was incorrect in my opinion. I expected to find trait means in the Tables, but only the results from the analysis (p-values) were shown! This is not acceptable. The actual means must be shown in Table 4. Too much emphasis was placed on the interactions. I would suggest presenting breed, sex and B*S interaction in Table 4 and leave age effect for the text or a Figure (growth curve). The superscripts in Table 5 do not make any sense Results continue to be presented in the Discussion section. Therefore, only a small proportion of the Discussion section is indeed Discussion. Part of the discussion is irrelevant.
An English language review is needed. There are typing mistakes and many grammar errors.
Statistical modeling seems adequate.
Specific comments:
I expected to find some main results (actual numbers, means and standard-errors) in the abstract. I understand that you have many traits, but show the most important ones.
Line 18: Why in the future? I believe there is enough interest today in alternative production systems (organic, free range) worldwide. Indigenous chicken breeds should be considered at least for crossbreeding in these systems.
Line 27: I suggest including “meat production and laying performance”.
Line 36: “The laying performances were lower for the two breeds,….”. What do you mean? Lower than?
Line 58: “Recently…”. Please provide references.
Line 62: Saying “intensive chicken breeds” is not correct. Suggestion: “intensively raised chicken breeds”.
Lines 76 – 86: Considering that these populations of these breeds are small, it is important to provide estimations of the population sizes in terms of males and females for both breeds.
Lines 95 – 96: Including a Figure with pictures of a male and a female from each breed would help the description.
Sample sizes for growth, carcass and egg production traits were small;
Lines 108, 109, 150 and 164: Use “scale” instead of “balance”.
Table 1: please organize this Table. The ingredients that appear in a larger proportion should appear first following a decreasing order.
Tittle of Tables 1, 3a and 3b must be revised. Table 1: delete the word “Ingredients”. Table 3a and 3b should be merged for consistency with Table 1. Use “Diet composition” in Tables 1 and 3.
The methods used in feed analysis should be indicated and properly described or referenced in the Material and methods section.
Line 134: use “were raised” instead of “were bred”.
Lines 134-135: Were the hens in individual cages from day 1? Please describe housing precisely in each phase.
Please number Tables in sequence (1, 2, 3, 4, etc.) avoiding 3a, 3b and 6a and 6b.
Line 161: Use “was used as the trait value”, instead of “was used as a trait”.
Line 167: “were”, not “was”.
The results and discussion section on growth performance and carcass traits is long, confusing and at the same time incomplete. The way the results are presented is incorrect in my opinion. I expected to find trait means in Table 4, but only the results from the analysis (p-values) were shown! This is not acceptable. The actual means must be shown in the Tables. Too much emphasis was placed on the interactions. I would suggest presenting breed, sex and B*S interaction in Table 4 and leave age effect for the text or a Figure (growth curve). Table 4 must be completely revised to contain the corrected means and standard errors for each trait and subclass.
Line 219: Use “between weeks 4 and 16” not “between 4 weeks and 16 weeks”.
Lines 231-232: It is not true. They should be, but they are not shown! All I see is asterisks and “NS”. Please revise this Table to contain the least-squares means and standard errors.
Lines 238-240: What do you mean? Please rewrite. It is unclear.
Line 253: Use “drip loss” (not losses) consistently.
Line 257: “several losses”. What do you mean? Please rephrase.
Line 265: The pH was higher in Ho than in Dong Tao only for pectoral meat.
Line 266: “thigh meat”, not “thigs meat”.
Line 269: delete “very significantly”. “Differed” followed by the p-value is enough.
Lines 269-279: cite Table 5.
The superscripts used in Table 5 do not make any sense to me. Please revise them thoroughly.
Table 6b should be Table 7.
Results continue to be presented in the Discussion section. Therefore, only a small proportion of the Discussion section is indeed Discussion. Part of the discussion is irrelevant. The entire discussion on growth performance and meat quality must be revised.
Line 327 – 334: this discussion on sexual dimorphism at one day of age is irrelevant. Please delete the entire paragraph.
Lines 363-364: “The native breeds have slower growth performance than the hybrid broilers [16] which might be due to poor feed conversion ratio”. Please delete this statement, it makes no sense. It could be the other way around, that is, feed conversion ratio is poor due to slow growth.
Line 406: “delicious sweetness”??? What do you mean”? Please explain.
Line 526: “small population size”, not “low size”.
This brings up an important issue: if the population sizes are too small in these breeds, you may have reached a genetic drift situation, with fixation of alleles at many loci. If this is the case, selection would not be effective to improve these populations.
Author Response
Dear reviewer,
first of all, we would like to thank you for reviewing our paper and for the helpful comments you provided. We have attempted to address all the point you raised as well as possible. We hope that you will find this new version of the paper better than the previous, and potentially suitable for publication in Animals!
Sincerely,
Duy Nguyen Van , Nassim Moula , Evelyne Moyse , Luc Do Duc , Ton Vu Dinh , Frederic Farnir

Reviewer 2 Report
Duy et al. used the Ho and Dong Tao chicken to verify the production performance and reproduction performance. The conclusion is that the production performances of Ho and Dong Tao chicken were poor.
The paper is not clearly presented. Some major revisions are required. In 3.1.1 and 3.1.2, the results of the growth, carcass yield and meat quality of each breed (LSM, SE) should be supplied as well instead of just show the comparison between two breeds.
Abstract
Line 38-29 This article only describes the slaughter performance, production performance and egg production performance of two chicken breeds. There have been no tests on how to improve the production and reproduction performance, so some explanations should be supplied.
Introduction
Line 67 Literature citation format is incorrect. Such as [8].
Materials and methods
Line 87 Is the font size of the first-level headings consistent with other headlines?
Such as, Materials and methods, Results.
Line 113-114 What are the conditions on which chickens are slaughtered every week? Chicken for selection in randomly? Or, selection by average weight?
Line 156-161 The sentence is malformed.
Results
Line 205-end The P-value is not capitalized. Why not show specific data on growth performance and slaughter performance?
Line 280 The expression of P-values is not clear, the P-value it is a weekly or all P-values?
Line 287 At the table 6a, what does “number of eggs per 52 weeks” mean?
Line 303 The sentence is malformed.
Discussion
Line 328 There should be spaces between numbers and letters.
Conclusions
The entire writing is not standardized and there are many language errors. The data that must be given is also not explained. the article format should be carefully revised.
Author Response

(The authors gave the same response as above.)

Reviewer 3 Report
refer to attached file

Author Response

(The authors gave the same response as above.)

Round 2
Reviewer 1 Report
Animals – 673633_v1
This article has the aim of describing growth, egg production and meat and quality traits of two local Vietnamese chicken breeds: Ho and Dong Tao.
This corrected manuscript is, to my view, clearly better than the previous version. The authors addressed most of the issues pointed out in the first review. However, some problems still remain and should be revised again.
The main issue now is an excess of Figures (there are 14!), many of which are unnecessary. I suggest omitting Figures 6 to 14. Although I agree that graphs are the best way to illustrate time series, one cannot show all the data this way.
In respect to the discussion, I think it has also been improved. However, I’d like to urge the authors to take into consideration the difference in population sizes between the two breeds for the discussion. I believe that some differences, such as in reproduction and survival indices, could be at least partially, attributed to inbreeding, which is closely related to population size. If you have a very small population, which seems to be the case for one of these breeds, genetic variation might have been reduced. Would selection for performance traits be effective in this situation? I am concerned about inbreeding and genetic drift (fixation at many loci) in the Ho population.
Another issue that should be clearly assessed refers to the real purpose of these breeds: are they used for meat production only (although you have measured egg production traits)?
An English language review is still needed. There are problems with verb tenses and concordance.
Specific comments:
Figures 1 and 2 were included in the manuscript, but they were not cited in the text. Please cite them when you describe the breeds either in the Introduction, or in Material and Methods or both.
The tittles of Figures 1 and 2 must be improved. For instance: Figure 1. Adult female (left) and male (right) Dong Tao chickens
The methods used in the analyses of hen’s feed were presented (lines 157-161). What about the feed for the growing chickens? Were they the same? (line 114).
In Table 3 “fiber” refers to crude fiber?
Omit Figures 6 to 14. Although I agree that graphs are the best way to illustrate time series, one cannot show all the data this way. Keep the description in the text.
Lines 248-251: “Neverthless…….from Dong Tao to Ho”. Delete this sentence. It was not significant, it is not worth mentioning.
Figure 5: What is the units for the y axis? It cannot be only grams. Grams of feed/ g of gain?
Figure 5: what do the error bars stand for? Standard-errors? If these bars are correct, differences could not be significant!
Do not use “significantly” repeatedly as in line 284. There is no need for that.
Lines 300-301: use “this trait”, not “these traits”;
Line 447: use “underlying”, not “below”;
Line 461: use “reproductive performance”, not “reproductive production”.
Author Response
Dear reviewer,
thanks a lot for reviewing our paper and for providing useful suggestions !
We have addressed your comments as well as possible, we hope our answers will satisfy you ! All answers are in the attached document.
Sincerely,

Reviewer 3 Report
see attached comments

Author Response
Dear reviewer,
thanks again for reviewing our work and providing useful suggestions !
We have answered to all your questions in the attached document, and we hope that these answers will be satifying !
Sincerely,

Round 3
Reviewer 3 Report
see attached comments

Author Response
Dear reviewer,
thanks for your useful comments and suggestions. You will find our answers in the attached document. We hope that you will consider our answers as acceptable.
Sincerely,
the authors
